# How many views does your deep neural network use for prediction?

## Abstract

The generalization ability of Deep Neural Networks (DNNs) is still not fully understood, despite numerous theoretical and empirical analyses. Recently, Allen-Zhu & Li (2023) introduced the concept of *multi-views* to explain the generalization ability of DNNs, but their main target is ensemble or distilled models, and no method for estimating multi-views used in a prediction of a specific input is discussed. In this paper, we propose *Minimal Sufficient Views (MSVs)*, which is similar to multi-views but can be efficiently computed for real images. MSVs is a set of minimal and distinct features in an input, each of which preserves a model's prediction for the input. We empirically show that there is a clear relationship between the number of MSVs and prediction accuracy across models, including convolutional and transformer models, suggesting that a multi-view like perspective is also important for understanding the generalization ability of (non-ensemble or non-distilled) DNNs.

## 1 Introduction

Deep Neural Networks (DNNs) perform very well on a variety of tasks, but the reason for the high generalization ability of DNNs has not yet been fully understood. Several approaches have been proposed to explain the generalization ability of DNNs theoretically or empirically, such as the neural tangent kernels (Jacot et al., 2018), the sharp minima discussion (Shrikumar et al., 2017), a PAC-Bayes perspective (Neyshabur et al., 2017), analysis for learning non-smooth functions (Sturmfels et al., 2020), memorization to random labeling (Zhang et al., 2017), to name a few.

Recently, Allen-Zhu & Li (2023) introduced the concept of *multi-views* for classification problems to explain the generalization performance of DNNs, especially in ensemble and distilled models. In multi-views, it is assumed that there are multiple features per class, and each input consists of a combination of them. For example, the "car" class is assumed to have several features, such as headlights, windows, and wheels, and each "car" image consists of some of these features. Allen-Zhu & Li (2023) then showed theoretically that the performance of DNN models can be improved by taking ensembles or distillation if the data has the multi-view structure. However, the relationship between the multi-view hypothesis and the generalization ability of standard (non-ensemble or non-distilled) DNN models is not discussed in (Allen-Zhu & Li, 2023). In addition, Allen-Zhu & Li (2023) do not address a method for estimating multi-views used in a prediction of a specific input.

In this paper, we propose *Minimal Sufficient Views (MSVs)* as a concept related to multi-views in (Allen-Zhu & Li, 2023), and propose a simple yet effective algorithm for estimating MSVs for real data. Our experiments suggest that a multi-view like perspective is also important for understanding the generalization ability of standard DNNs, including convolution or transformer models, not only specifically for ensemble or distilled models as discussed in Allen-Zhu & Li (2023). Given a prediction model and an input to the model, we define MSVs as a set of minimal and distinct features in the input, such that each feature preserves the model's prediction for the input unchanged. We empirically show that a standard (non-ensemble or non-distilled) DNN model uses multiple views in its prediction, in the sense of MSVs. Interestingly, the number of views used in a prediction varies depending on the model and input, and there is a clear relationship between the number of views used in the prediction and the accuracy of the prediction: the more views the prediction uses, the more accurate the prediction is. In addition, when we have a set of DNN models trained on the same dataset, the model with the larger average number of MSVs achieves better generalization perfor-

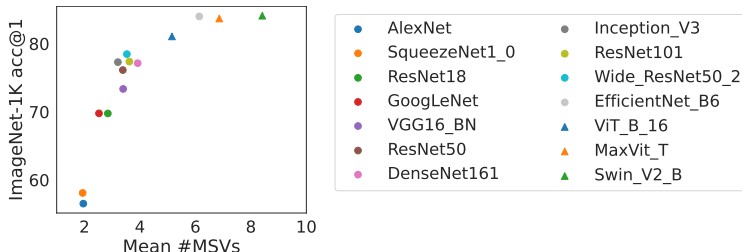

Figure 1: Average number of estimated MSVs for randomly sampled 1000 images from the ImageNet validation set (x-axis) and prediction accuracy on all data in the ImageNet validation set (y-axis). Note that no label information is used to compute the MSVs. ImageNet-trained DNN models obtained from TorchVision are used in the evaluation (listed in the legend). See Section 4 for the experimental details.

mance, as shown in Figure 1. This result suggests that one reason for the high performance of DNN models is that their prediction is supported by multiple views of an input. Since label information is *not* required to compute MSVs, Figure 1 has a practical implication that we can select the best performing DNN model by evaluating the proposed MSVs on unlabeled data.

The proposed MSVs can also be adopted as a method for XAI (eXplainable AI), since it shows important parts of a given input from the viewpoint of preserving the prediction for the image. In contrast to existing XAI methods that assume something like 1-view, i.e., a single heatmap is used to explain the prediction, we can understand the prediction from multiple views with MSVs. Our experiments suggest that it would be insufficient to explain the prediction of a highly nonlinear DNN with a single heatmap, especially when the prediction is based on multiple views.

The rest of the paper is organized as follows: In Section 2, we review existing work related to MSVs. In Section 3, we first formally define MSVs, and then propose a greedy algorithm to estimate MSVs. We give empirical results in Section 4 and summarize the work in Section 5.

## 2 RELATED WORK

Allen-Zhu & Li (2023) introduced the *multi-view structure* in data for $K$-class classification problems. Here we briefly review the concept of multi-view: In multi-view, it is assumed that for each class $k \in [\![K]\!]$[1] there are $J_k$ latent features[2] $\{v_{k,j} \in \mathbb{R}^d \mid j \in [\![J_k]\!]\}$. An input $x$ in a dataset is assumed to consist of $P$ vectors, each of which is a linear combination of $v_{k,j}$. When the class of $x$ is $k'$, the coefficients of $v_{k',j}$ have relatively high values. Under the multi-view assumption, Allen-Zhu & Li (2023) theoretically showed that the generalization performance of DNNs is improved by taking ensembles or distillation. As evidence of a trained model using multi-views, they showed synthesized images generated to maximize the activation of specific neurons in the model, but it is not discussed how to estimate the multi-views used in a prediction for a specific model and input. In contrast to Allen-Zhu & Li (2023), which assumes common latent features across inputs, the MSVs proposed in this paper is defined on a single input and model, and thus has no direct relation to the multi-views in (Allen-Zhu & Li, 2023). However, as our experiments show, the MSVs obtained from images of the same prediction class tend to have shared features, e.g., images predicted to be of the cat class tend to have MSVs indicating cat eyes, even though they are estimated independently from each image, suggesting the relationship between MSVs and multi-views in (Allen-Zhu & Li, 2023). In addition, we explore the relationship between the number of MSVs and the prediction accuracy, such a discussion is not included in (Allen-Zhu & Li, 2023).

There has been a large amount of research on XAI. One of the main approaches in XAI research is to help interpret the prediction result of a DNN model for a given input by showing the important parts of the image for the prediction (Selvaraju et al., 2017; Ribeiro et al., 2016; Sundararajan et al., 2017; Shrikumar et al., 2017; Fong & Vedaldi, 2017; Petsiuk et al., 2018). However, all of these methods display their analysis results in a single heatmap, which is equivalent to assuming a single

---

[1]For a positive integer $A$, we define $[\![A]\!] := \{1, \ldots, A\}$.
[2]$J_k = 2$ is assumed in (Allen-Zhu & Li, 2023) for simplicity.

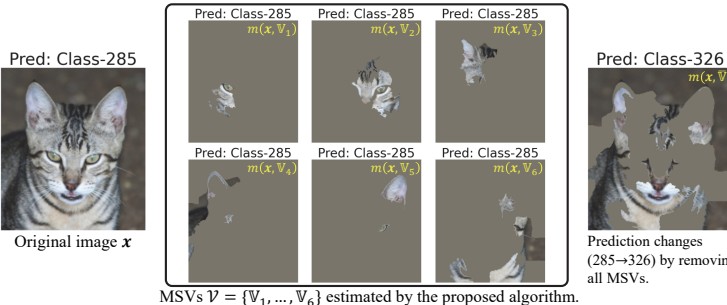

Figure 2: Example of estimated MSVs for an image in the Open Images validation set. See Section 3.2 for definitions of mathematical expressions.

view in the prediction. Carter et al. (2019) proposed SIS, which considers the multiple views similar to our definition, but its relationship to the generalization performance is not discussed. From an algorithmic point of view, our method improves computational efficiency by relaxing the minimality condition compared to Carter et al. (2019). Carter et al. (2021) proposed a method to improve the efficiency of SIS, but it requires the computation of gradients, avoiding the application to black-box models. Counterfactual is another line of research for XAI (Wachter et al., 2018; Mothilal et al., 2020). Given a model and an input, these approaches attempt to find (or generate) another input, called the counterfactual, that is as close as possible to the original input, while the predicted outcome is different from the original. The prediction is then interpreted by showing the difference between the original input and its counterfactual. In contrast, the proposed algorithm tries to find the smallest part of the input that preserves the prediction. There have been several attempts to learn DNN models that are intrinsically interpretable. Prototype-based models (Chen et al., 2019; Nauta et al., 2021) are one of them, where heatmaps showing the similarity between the given image and the trained prototypes are presented along with the prediction. Unlike these approaches, our method can be applied to any DNN model, requiring only the input-output relationship of the model.

## 3 PROPOSED METHOD

### 3.1 EXAMPLE OF ESTIMATED MSVS

Before moving on to the formal definitions of the proposed MSVs, we first show a specific example to facilitate understanding. Figure 2 shows an example of MSVs obtained using the algorithm we will describe in Section 3.3. The left side of Figure 2 depicts an image[3] from the validation split of the Open Images dataset (Kuznetsova et al., 2020). We applied a ResNet-101 model (He et al., 2016) trained with the ImageNet dataset (Deng et al., 2009) to the image, and obtained the predicted class of 285 (*Egyptian Cat*) out of 1000 classes of ImageNet. The center of Figure 2 shows the MSVs estimated by the proposed algorithm with respect to the ResNet-101 prediction, which consists of 6 images with different masked patterns. Note that all of these masked images are still classified as Class 285 by the ResNet, indicating that there are 6 views in the prediction for the image, each of which is sufficient to make the prediction that the image is *Egyptian Cat*. We can see from these MSVs that the prediction for the left image is supported by several features, including the right eye and the face stripe pattern, the left eye and the face stripe pattern, the right ear, the left ear, and the body stripe pattern. The right side of Figure 2 shows the image with all MSVs inverted, resulting in the prediction change from 285 to 326 (*Lycaenid*).

### 3.2 MINIMAL SUFFICIENT VIEWS

Now we give a formal definition of the proposed MSVs. For simplicity, the following definitions are for vector inputs, but the same definitions can be applied to images and other inputs. Given a $K$-class classification problem, let the input be $x \in \mathbb{R}^n$ and the prediction model be $f : \mathbb{R}^n \to \mathbb{R}^K$.

---

[3]To clarify the licenses, we have used images from the Open Images dataset when illustrating the proposed method or other methods using specific images. See Appendix J for the URLs of the original images.

The $i$-th element of the vector $\boldsymbol{x}$ is denoted by $x_i$ and the $j$-th output of the multivalued function $f$ is denoted by $f_j$. Let $c_f(\boldsymbol{x}) := \arg\max_{k\in[\![K]\!]} f_k(\boldsymbol{x})$ be the function that determines the prediction class for $\boldsymbol{x}$ based on $f$.

We denote a *view* for an input $\boldsymbol{x}$ as $\mathbb{V} \subseteq [\![n]\!]$ $(\mathbb{V} \neq \emptyset)$, that is, a subset of the indices of the elements of $\boldsymbol{x}$. The masked input $m(\boldsymbol{x}, \mathbb{V})$ for an input $\boldsymbol{x}$ and a view $\mathbb{V}$ is then defined as follows:

$$m(\boldsymbol{x}, \mathbb{V}) := (m_1, \ldots, m_n)^\top, \quad m_i = \left\{ \begin{array}{ll} x_i & \text{if } i \in \mathbb{V}, \\ b_i & \text{otherwise,} \end{array} \right. \quad \text{for } i \in [\![n]\!],$$

where $\boldsymbol{b} \in \mathbb{R}^n$ is the baseline input, which is also used in existing XAI methods, including integrated gradients (Sundararajan et al., 2017) or DeepLift[4] (Shrikumar et al., 2017). There is no single way to determine the baseline input, and there are various options as discussed in Sturmfels et al. (2020). In this paper, we use the average value of $\boldsymbol{x}$ in the training data as $\boldsymbol{b}$. Examples of masked inputs are shown in the middle of Figure 2, in which the gray area corresponds to the baseline $\boldsymbol{b}$.

**Definition 1.** Given $f$, a view $\mathbb{V}$ for $\boldsymbol{x}$ is *minimal sufficient* if:

$$\begin{array}{ll} \text{(Sufficiency)} & c_f(\boldsymbol{x}) = c_f(m(\boldsymbol{x}, \mathbb{V})), \\ \text{(Minimality)} & c_f(\boldsymbol{x}) \neq c_f(m(\boldsymbol{x}, \mathbb{V} \setminus \{i\})), \quad \forall i \in [\![n]\!] \setminus \mathbb{V}. \end{array}$$

By definition, sufficiency means that the prediction of $f$ for $\boldsymbol{x}$ is unchanged by applying the view, while minimality means that the prediction is changed after removing an element from the view. We now give our definition of MSVs:

**Definition 2.** Given $f$, *Minimal Sufficient Views (MSVs)* $\mathcal{V}$ for $\boldsymbol{x}$ is:

$$\mathcal{V} = \{\mathbb{V}_1, \ldots, \mathbb{V}_q\} \text{ such that } \mathbb{V}_i \text{ is minimal sufficient } \forall i \in [\![q]\!], \mathbb{V}_i \cap \mathbb{V}_j = \emptyset \text{ if } i \neq j,$$

$$\text{and } c_f(\boldsymbol{x}) \neq c_f(m(\boldsymbol{x}, \bar{\mathbb{V}})), \text{ where } \bar{\mathbb{V}} = [\![n]\!] \setminus \bigcup_{i \in [\![q]\!]} \mathbb{V}_i.$$

By definition, MSVs is a set of disjoint views, each of which is minimal sufficient, as shown in the middle of Figure 2. The last condition indicates that $\bar{\mathbb{V}}$ is not sufficient, i.e. excluding all views in the MSVs changes the prediction, as shown on the right side of Figure 2. Note that MSVs are not necessarily unique for given $f$ and $\boldsymbol{x}$.

### 3.3 Greedy Algorithm

It is computationally expensive to find exact MSVs, especially when the input dimension $n$ is large. We propose a greedy algorithm to heuristically compute MSVs, in which Minimality condition in Definition 1 is relaxed to $\beta$-Split-Minimality.

$\beta$**-Split-Minimality.** First we introduce the Split function that splits a view $\mathbb{V}$, which is an index set, into $\beta$ groups using $\boldsymbol{x}$ as an auxiliary. If $\boldsymbol{x}$ is an image, superpixel methods such as SLIC (Achanta et al., 2012) can be used as Split, and the Voronoi partition (de Berg et al., 2000) can also be used as an $\boldsymbol{x}$-independent splitting method. We can speed up the search for MSVs by exploring the group units that are obtained by Split instead of exploring each element of $\boldsymbol{x}$. Note that Split will split $\mathbb{V}$ into $\beta' = |\mathbb{V}|$ groups if $|\mathbb{V}| < \beta$.

We define the $\beta$-Split-Minimality using Split as follows:

$$(\beta\text{-Split-Minimality}) \quad c_f(\boldsymbol{x}) \neq c_f(m(\boldsymbol{x}, \mathbb{V} \setminus \mathbb{S})), \quad \forall \mathbb{S} \in \text{Split}(\mathbb{V}, \boldsymbol{x}, \beta)$$

By setting $\beta$ sufficiently large $(\beta = |\mathbb{V}|)$, $\beta$-Split-Minimality becomes equivalent to Minimality condition in Definition 1, but the larger $\beta$ is, the higher the computational cost of the following GreedyMSVs. Our experiments show that a relatively small $\beta$ (4 to 32) is enough to obtain the results corresponding to Figure 1.

**GreedyMSVs.** We propose GreedyMSVs in Algorithm 1 to compute MSVs that satisfy $\beta$-Split-Minimality. In Algorithm 1, the submodule EstimateMSV is called recursively, and each time it is called, the view at that point is divided into $\beta$ groups by Split. In this way, in the early stages of the search, when the view size is large, the search can be performed with a coarse partition. As the search progresses and the view size gets smaller, the search can be performed with a more detailed partition. See the execution example in the next section for details. Every view in $\mathcal{V}$ obtained by Algorithm 1 satisfies both Sufficiency and $\beta$-Split-Minimality.

---

[4]It is called "reference input" in DeepLift (Shrikumar et al., 2017).

---

**Algorithm 1** Greedy algorithm to compute MSVs

---

**[Input]** $\boldsymbol{x}$: input to the model, $f$: classification model
**[Parameter]** $\beta$: number of splits
**[Output]** $\mathcal{V}$: MSVs

1: **function** GREEDYMSVS($\boldsymbol{x}, f$)
2:     $\mathcal{V} \leftarrow \emptyset, \mathbb{V}_0 \leftarrow [\![n]\!]$                ▷ Initialize
3:     $k \leftarrow c_f(\boldsymbol{x})$                ▷ Predicted class for $\boldsymbol{x}$
4:     **do**
5:         $\mathbb{V} \leftarrow$ ESTIMATEMSV($\boldsymbol{x}, f, \mathbb{V}_0, k$)          ▷ Estimate an MSV
6:         $\mathcal{V} \leftarrow \mathcal{V} \cup \{\mathbb{V}\}$             ▷ Add to $\mathcal{V}$
7:         $\mathbb{V}_0 \leftarrow \mathbb{V}_0 \setminus \mathbb{V}$            ▷ Remove used indexes
8:     **while** $c_f(m(\boldsymbol{x}, \mathbb{V}_0)) = k$       ▷ Continue as long as $\mathbb{V}_0$ is sufficient
9:     **return** $\mathcal{V}$
10: **function** ESTIMATEMSV($\boldsymbol{x}, f, \mathbb{V}, k$)
11:     $\mathcal{S} \leftarrow$ SPLIT($\mathbb{V}, \boldsymbol{x}, \beta$)         ▷ Split indexes into groups
12:     $\mathbb{S}' \leftarrow \arg\min_{\mathbb{S} \in \mathcal{S}} |f_k(\boldsymbol{x}) - f_k(m(\boldsymbol{x}, \mathbb{V} \setminus \mathbb{S}))|$    ▷ Select $\mathbb{S}'$ with minimal change
13:     $\mathbb{V}' \leftarrow \mathbb{V} \setminus \mathbb{S}'$
14:     **if** $c_f(m(\boldsymbol{x}, \mathbb{V}')) = k$ and $|\mathbb{V}'| > 0$ **then**   ▷ Judge whether $\mathbb{V}'$ is sufficient or not
15:         $\mathbb{V} \leftarrow$ ESTIMATEMSV($\boldsymbol{x}, f, \mathbb{V}', k$)       ▷ Recursively shrink $\mathbb{V}$
16:     **return** $\mathbb{V}$

---

Figure 3: Execution example of MSVs search by GREEDYMSVS.

**Execution Example** Figure 3 shows an example of searching for MSVs with GREEDYMSVS. We used an image from the Open Images validation set and a ResNet-101 model trained with ImageNet. We used SLIC as SPLIT. The yellow line in the figure shows the partition obtained by SPLIT with $\beta = 16$. The upper left image shows the original image classified by the model as *Tiger Shark*, and the next image to the right shows the result by SPLIT and $\mathbb{S}'$ is the region selected in Line 12 of Algorithm 1, which minimally changes the prediction. Then $\mathbb{S}'$ is removed from the current view and SPLIT is applied recursively. We eventually get the first MSV $\mathbb{V}_1$, where the prediction changes if we remove any region obtained by SPLIT. As can be seen in the figure, the recursive application of SPLIT results in a gradual refinement of the resulting partition, allowing a fine-grained adaptive search of a view while speeding up the search. Then MSV $\mathbb{V}_1$ is removed from $\mathbb{V}_0$ and the search continues as long as the updated $\mathbb{V}_0$ is sufficient, i.e. $m(\boldsymbol{x}, \mathbb{V}_0)$ is classified as *Tiger Shark*.

## 4 EXPERIMENTAL RESULTS

### 4.1 SETUP FOR CLASSIFICATION

Otherwise unless stated, we used SLIC implemented in (van der Walt et al., 2014) as SPLIT in the proposed algorithm with $\beta = 16$. We used ImageNet (Deng et al., 2009), Open Images (Kuznetsova

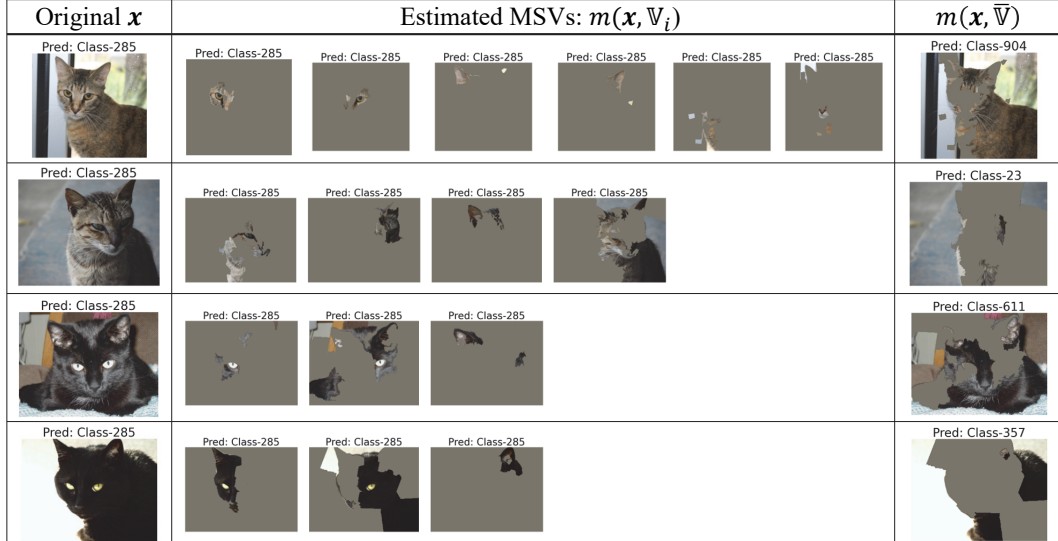

Figure 4: MSVs for images from the Open Images validation set that are predicted to be Class 285.

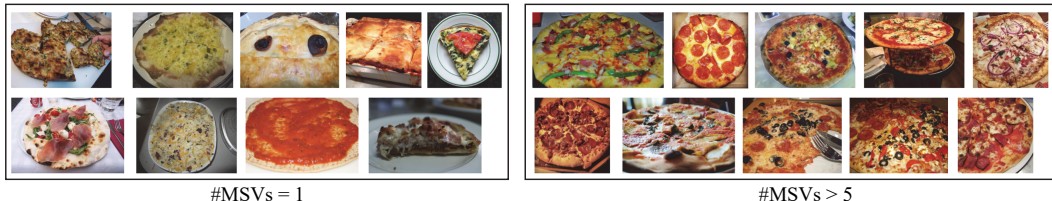

Figure 5: Images from the Open Images validation set predicted to be Class 963. The number of estimated MSVs is 1 for the left group and greater than 5 for the right group.

et al., 2020), ImageNet-C (Hendrycks & Dietterich, 2019), or CIFAR-100 (Krizhevsky & Hinton, 2009) as the dataset for image classification. We obtained pretrained classification models from TorchVision[5], which are trained with the ImageNet training set, and used them for evaluation.

### 4.2 MSV COMPARISON BETWEEN IMAGES OF THE SAME PREDICTION CLASS

We computed MSVs for images from the Open Images validation dataset using ResNet-101, an example of which was shown in Figure 2. Figure 4 shows additional results where we computed MSVs for other images predicted to be Class 285 (*Egyptian Cat*). Interestingly, even though MSVs were obtained separately for each image, MSVs with common features were obtained for multiple images, such as the left and right eyes and the left and right ears, which is consistent with the multi-view in Allen-Zhu & Li (2023) that assumes common latent features across images. In Appendix A, we give estimated MSVs for other images that also share common features across images in the same prediction class. Note that there is no clear trend in the prediction change after removing the estimated MSVs (see the prediction class for $m(\boldsymbol{x}, \overline{\mathbb{V}})$ in the figure). We can see that the number of estimated MSVs varies depending on the image, ranging from 3 to 6 in Figure 2 and Figure 4.

### 4.3 COMPARING IMAGES BY ESTIMATED NUMBER OF MSVs

We computed MSVs for images from the Open Images dataset predicted by ResNet-101 to be Class 963 (*Pizza*). The left side of Figure 5 shows images with $|\mathcal{V}| = 1$, where $\mathcal{V}$ is the estimated MSVs, with respect to the prediction to be Class 963, while the right side of the figure shows images with $|\mathcal{V}| > 5$. We see that the right group includes typical pizza images with tomato sauce

---

[5]https://pytorch.org/vision/stable/models.html (accessed August 2, 2023)

Table 1: Prediction accuracy for the ImageNet validation set, in which images are grouped based on the number of estimated MSVs.

| #MSVs | 1 | 2 | 3 | 4 | 5 | 6 | 7 | 8 | 9 | $\geq 10$ |
|---|---|---|---|---|---|---|---|---|---|---|
| Accuracy | 0.538 | 0.808 | 0.865 | 0.891 | 0.905 | 0.919 | 0.919 | 0.922 | 0.923 | 0.937 |

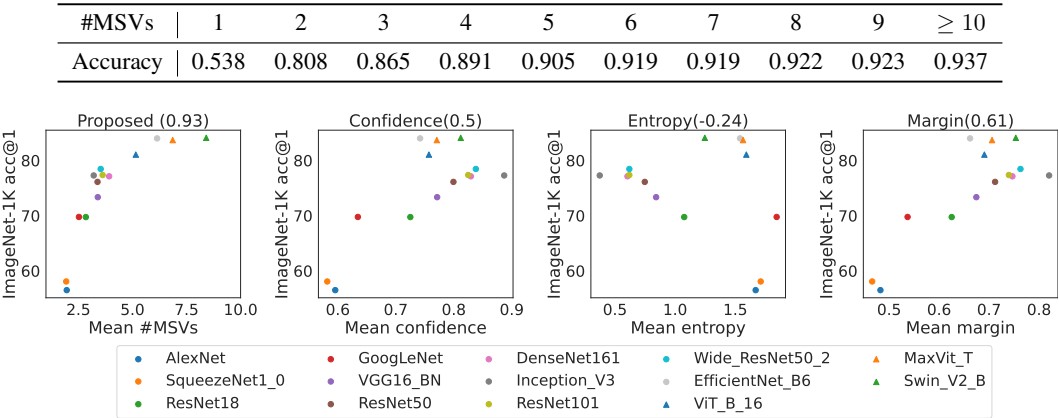

Figure 6: Average score versus top 1 accuracy. Values in parentheses for each method are rank correlations between score and accuracy.

and cheese with toppings like salami or olives. In contrast, the left group contains pizza images without tomato sauce or toppings, or even non-pizza images. These results suggest that the number of estimated MSVs is related to the difficulty of the prediction. In fact, we show in the next section that the prediction accuracy is clearly related to the number of estimated MSVs. Furthermore, if we have two prediction models trained on the same dataset, the model that uses more MSVs in its prediction has a higher generalization ability, which will be discussed in Section 4.5.

## 4.4 PREDICTION ACCURACY VS. ESTIMATED NUMBER OF MSVS

Using ResNet-101, we classified each image in the ImageNet validation set, consisting of 50000 images, and computed MSVs in each prediction. Note that MSVs are calculated based on the predicted classes, so information about the true class label in the validation set is *not* used to calculate MSVs. We grouped the validation images according to the number of estimated MSVs and evaluated the prediction accuracy in each group, the results of which are shown in Table 1. Interestingly, there is a clear relationship between the number of estimated MSVs and the accuracy of the prediction: the more MSVs the prediction is based on, the more accurate the prediction is.

## 4.5 MODEL COMPARISON USING AVERAGE NUMBER OF MSVS

To further investigate the relationship between the number of MSVs and generalization ability, we computed MSVs for various ImageNet-trained models obtained from TorchVision, which are listed in the legend of Figure 6, including convolutional neural networks and transformer-based models. We randomly sampled 1000 images from the ImageNet validation dataset and computed MSVs for these images using each model. One reason for using a subset of the full validation set is to reduce computation time, but as our results suggest, it is sufficient to use a subset to evaluate models using MSVs. The leftmost figure in Figure 6, which was also shown in Figure 1, shows the relationship between the average number of MSVs over these 1000 images (x-axis) and the prediction accuracy over the entire validation set (y-axis) of each model. We can see a clear relationship that a model with a larger number of MSVs achieves higher prediction accuracy. Note that the number in parentheses in the figure indicates a rank correlation between the score on the x-axis and the accuracy on the y-axis, which is as high as 0.93 for the proposed MSVs. As mentioned above, since the calculation of MSVs does not require true labels, these results suggest that it is possible to select a model by evaluating the MSVs of an unlabeled dataset. In addition, the average number of MSVs computed using the training dataset itself can also be used for model selection, as we will discuss later.

**Comarison with other metrics.** For each model, we also computed metrics representing prediction uncertainty, including confidence, entropy, and 1-vs-2 margin Settles (2009), using the same

1000 images from the ImageNet validation set. Note that these metrics can also be computed without using label information. It is expected that a model's prediction will be more accurate if the prediction uncertainty is low on average, provided that the model is well calibrated (Guo et al., 2017). The results are summarized in the rest of Figure 6, where the x-axis represents the average of each metric over 1000 images, and the rank correlations between score and accuracy are shown in parentheses. We see that all metrics have lower correlations than the proposed MSVs. Confidence and margin have a high correlation with accuracy when only convolutional models, except for EfficientNet_B6, are considered, but it decreases when transformer models, marked by triangles in the figure, are included. In contrast, MSVs show a consistent relationship in both models.

**Assessing the effect of $\beta$.** The proposed algorithm for estimating MSVs has a parameter $\beta$, which was $\beta = 16$ in previous sections, that controls the number of groups obtained by SPLIT. To investigate the impact of $\beta$, we computed MSVs with $\beta$ varying from 4 to 32, with other experimental settings the same as in the previous section. The results are shown in Figure 12 in Appendix B, with the rank correlations in parentheses. From Figure 12, we can see that the average number of MSVs has a high correlation with accuracy under fixed $\beta$, regardless of its value. We see that the average number of MSVs decreases as $\beta$ is set smaller. This is natural because the size of each MSV tends to be large as $\beta$ gets smaller, since the $\beta$-Split-Minimality becomes harder to hold.

**MSVs estimation using training dataset.** Instead of using images from the ImageNet *validation* set, we evaluated each score, including MSVs and uncertainty scores, using 1000 images that are randomly sampled from the ImageNet *training* set. Figure 13 in Appendix C shows the comparison between the estimated scores, where the x-axis is the score estimated using the training set and the y-axis is the score estimated using the validation set. We see that there are gaps between the scores estimated using the training and validation sets in confidence, entropy, and margin. This is because models tend to be overconfident for the data used in training. In contrast, the gap between the scores estimated using the training and validation sets is much smaller for the average number of MSVs, suggesting that we can use the data used for training itself to evaluate the average number of MSVs for comparing models. Such a property is extremely useful because it eliminates the need to prepare holdout data when training and selecting models.

**Evaluation with other classification datasets.** We computed MSVs and other uncertainty scores using the CIFAR-100 validation set, in which we used models that are obtained by finetuning models from TorchVision using the CIFAR-100 training set. The results are shown in Figure 14 in Appendix D, showing the advantage of MSVs against other uncertainty metrics. We also evaluated MSVs for the ImageNet-C dataset (Hendrycks & Dietterich, 2019), which consists of 75 datasets for evaluation that are created by distorting images in ImageNet with different types (15 types) and levels (5 levels) of distortion. In the experiment with ImageNet-C, we used ResNet-50 and ViT_B_16 as representatives of convolutional and transformer-based models, respectively, to reduce the computational burden. The results are shown in Figure 15 in Appendix E, in which we can see that the rank correlation between the average number of estimated MSVs and the prediction accuracy is still as high as 0.92 in the ImageNet-C evaluation.

### 4.6 COMPARISON WITH EXISTING XAI METHODS

As shown in Figure 2, the proposed MSVs allow us to understand which part of a given image is sufficient to maintain the prediction results of a given classification model. We can use MSVs as an XAI method to investigate how the given model makes its prediction. For example, from Figure 2, we can see that the model makes its prediction that the image is an *Egyptian Cat* from multiple views, such as the right eye and the face stripe pattern, the left eye and the face stripe pattern, the right ear, the left ear, and the body stripe pattern. To compare MSVs with other XAI methods, we applied existing XAI methods, including LIME (Ribeiro et al., 2016), Integrated Gradients (Sundararajan et al., 2017), GradCAM (Selvaraju et al., 2017), Guided-GradCAM (Selvaraju et al., 2017), and Occlusion (Zeiler & Fergus, 2014), to the prediction of the image in Figure 2. We used the author's implementation for LIME and Captum (Kokhlikyan et al., 2020) for other methods. Figure 7 shows the results obtained with these methods[6]. We see that the visualization results vary depending on the

---

[6]We used the default parameters of each library. To apply GradCAM, we used the last layer in the 4th residual layer of ResNet-101.

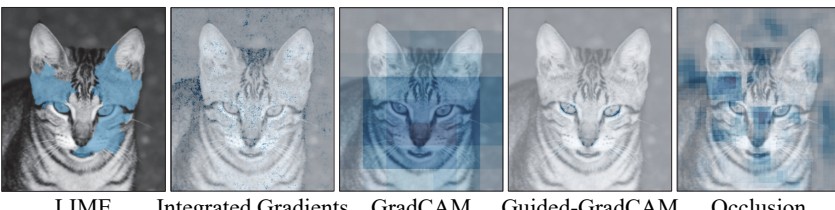

Figure 7: Results of applying other XAI methods to the prediction in Figure 1.

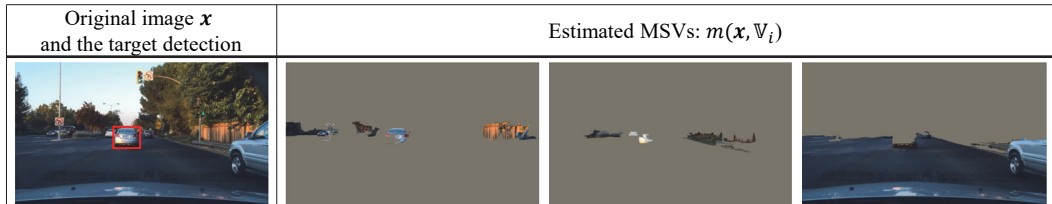

Figure 8: Estimated MSVs for the detection in the red box in the left image.

method used: Some highlight the eyes, some the nose, and some the ears. One possible reason for such an inconsistent result is that the model's prediction is made from multiple views, as our MSVs suggest. It would be insufficient to explain the prediction of a highly nonlinear DNN from a single viewpoint, such as a single heatmap. In contrast, the proposed MSVs can investigate the prediction of DNNs by explicitly considering the multiple views.

### 4.7 APPLICATION TO A DETECTION MODEL

We computed MSVs for a detection model, for which we used YOLOP (Wu et al., 2022) trained on the BDD100K dataset (Yu et al., 2020). We need to define $f$ properly to apply Algorithm 1. The details of customizing $f$ for a detection model are described in Appendix F. Figure 8 shows an example of computed MSVs for an image from the BDD100K validation set, where we computed MSVs for the detection marked by the red box in the leftmost image. From Figure 8, we can see that the detection is supported by three views: the rear lights and the rear window of the center car, a part of the rear of the car and the car next to it, and the bottom part of the car and the road. As this example shows, we can apply MSVs to a variety of models by properly setting $f$ in Algorithm 1.

## 5 CONCLUSION AND FUTURE WORK

In this paper, we proposed MSVs as a set of minimal and distinct features in the input, such that each feature preserves the model's prediction for the input, and proposed a greedy algorithm to efficiently compute approximate MSVs. Our experiments suggest that the number of MSVs corresponds to the prediction accuracy, which holds for various models, including convolution and transformer models. Since the computation of MSVs does not require labels, it is suggested that we can select the best performing DNN model by evaluating the average MSVs on unlabeled data. The proposed MSVs can be used as an XAI method that can explain the prediction of DNNs from multiple views. There are several directions in which this work can be extended. The first direction is to apply MSVs to prediction models whose inputs are not images, such as language models. We believe that MSVs have a wide range of applications because it only requires the input-output relationship of the model, i.e., the model can be a black box. The second direction is to apply MSVs to active learning. Since we can compute MSVs for unlabeled datasets, it is expected that we can find some kind of hard examples from unlabeled datasets by selecting images that have a small number of MSVs.

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

## A  EXAMPLES OF ESTIMATED MSVS

We computed MSVs for images from the Open Images validation dataset using TorchVision's ImageNet-trained ResNet-101. Figures 9-11 show results including images predicted by ResNet-101 to be classes 301 (*Ladybug*), 24 (*Great grey owl*), and 339 (*Sorrel*), respectively. We can see that MSVs with common features are obtained from multiple images predicted to be of the same class. For example, images predicted to be *Ladybug* have MSVs that contain black dots in a red pattern. For images predicted to be *Great grey owl*, MSVs show the right or left side of the face as a common pattern. For images predicted to be *Sorrel*, part of a leg with muscles are estimated as shared MSVs. It is interesting to observe that such MSVs with common features are estimated even though they are estimated separately for each image, suggesting the relationship between MSVs and multi-view in Allen-Zhu & Li (2023).

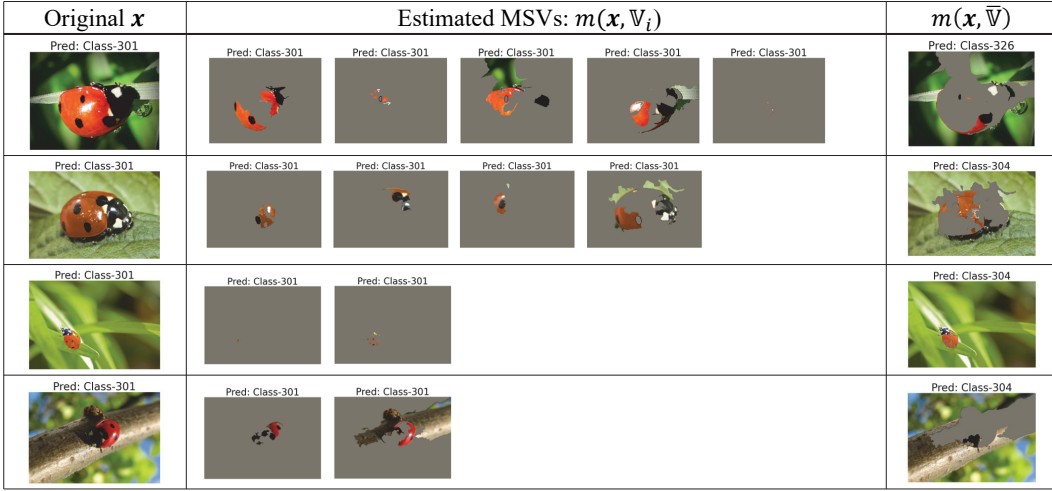

| Original $\boldsymbol{x}$ | Estimated MSVs: $m(\boldsymbol{x}, \mathbb{V}_i)$ | | | | | $m(\boldsymbol{x}, \overline{\mathbb{V}})$ |
|---|---|---|---|---|---|---|
| Pred: Class-301 | Pred: Class-301 | Pred: Class-301 | Pred: Class-301 | Pred: Class-301 | Pred: Class-301 | Pred: Class-326 |
| Pred: Class-301 | Pred: Class-301 | Pred: Class-301 | Pred: Class-301 | Pred: Class-301 | | Pred: Class-304 |
| Pred: Class-301 | Pred: Class-301 | Pred: Class-301 | | | | Pred: Class-304 |
| Pred: Class-301 | Pred: Class-301 | Pred: Class-301 | | | | Pred: Class-304 |

Figure 9: MSVs for images from the Open Images validation set that are predicted to be Class 301 by ResNet-101.

| Original $\boldsymbol{x}$ | Estimated MSVs: $m(\boldsymbol{x}, \mathbb{V}_i)$ | | | | $m(\boldsymbol{x}, \overline{\mathbb{V}})$ |
|---|---|---|---|---|---|
| Pred: Class-24 | Pred: Class-24 | Pred: Class-24 | Pred: Class-24 | Pred: Class-24 | Pred: Class-971 |
| Pred: Class-24 | Pred: Class-24 | Pred: Class-24 | Pred: Class-24 | | Pred: Class-63 |
| Pred: Class-24 | Pred: Class-24 | Pred: Class-24 | | | Pred: Class-146 |
| Pred: Class-24 | Pred: Class-24 | Pred: Class-24 | | | Pred: Class-326 |

Figure 10: MSVs for images from the Open Images validation set that are predicted to be Class 24 by ResNet-101.

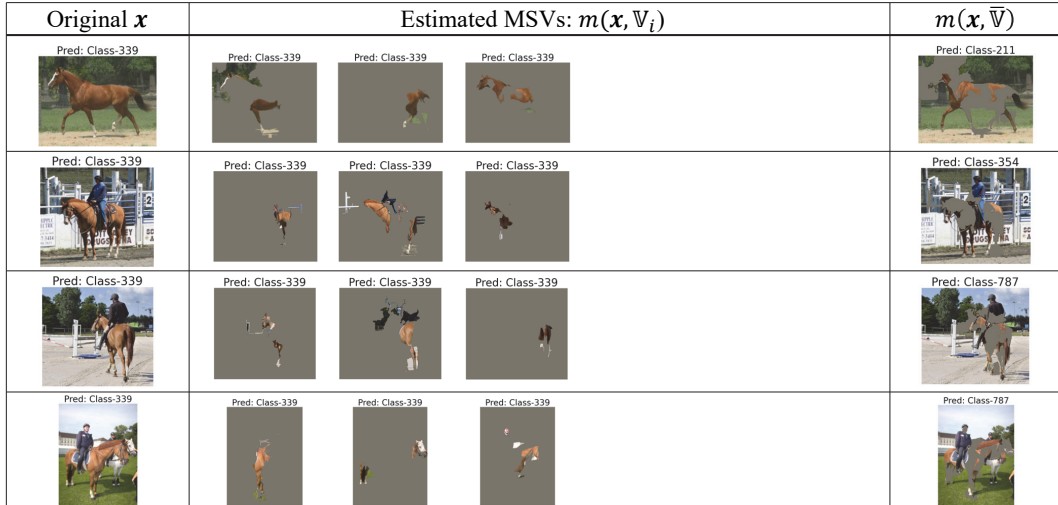

Figure 11: MSVs for images from the Open Images validation set that are predicted to be Class 339 by ResNet-101.

## B    ASSESSING THE EFFECT OF $\beta$

We computed MSVs using Algorithm 1 with $\beta$ varying from 4 to 32 for TorchVision's ImageNet-trained models using randomly sampled 1000 images from the ImageNet validation set. Figure 12 shows the results, where the x-axis represents the average number of MSVs over these 1000 images and the y-axis represents the prediction accuracy over the entire ImageNet validation set, with the rank correlations in parentheses. We can see that the average number of MSVs has a high correlation with accuracy under fixed $\beta$, regardless of its value.

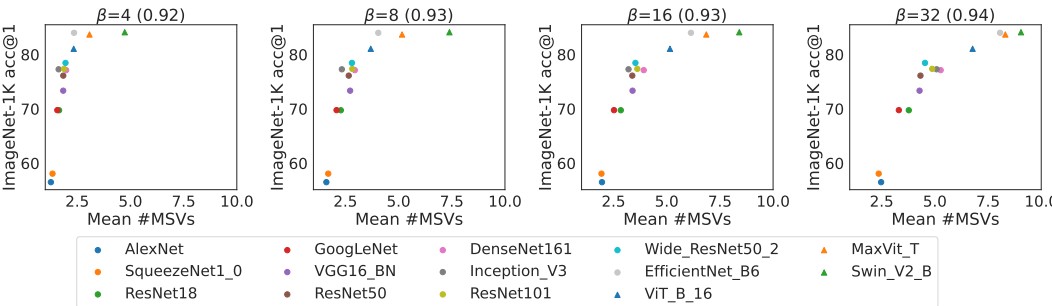

Figure 12: Average number of MSVs versus top 1 accuracy with $\beta$ changing from 4 to 32. Values in parentheses are rank correlations.

### B.1    COMPUTATION TIME

We evaluated the computation time of Algorithm 1 with ResNet-101 using 1000 images sampled from the ImageNet validation set used in the evaluation of Figure 12. The experiment was performed on a single machine with a single GPU (Nvidia A100). The results are summarized in Table 2, in which the computation time is averaged over 1000 images. We can see that the computation time decreases as we use smaller $\beta$. From the table, it will take about less than 8 minutes to compute the MSVs for these 1000 images when $\beta = 4$. Since MSVs can be computed in parallel with respect to each image, we can further accelerate the computation of MSVs by applying parallel computing.

Table 2: Computation time of estimating MSVs with Algorithm 1.

| $\beta$ | 4 | 8 | 16 | 32 |
|---|---|---|---|---|
| Comp. time per image (sec) | 0.46 | 1.97 | 8.48 | 41.56 |

## C    MSVs ESTIMATION USING TRAINING DATASET

Instead of using images from the ImageNet *validation* set, we evaluated each score, including the average number of MSVs and other uncertainty scores, using 1000 images that are randomly sampled from the ImageNet *training* set. Figure 13 shows the comparison between the estimated scores, where the x-axis is the score estimated using the training set and the y-axis is the score estimated using the validation set. We can see that the average number of MSVs has a smaller gap between the scores estimated using the training and validation sets compared to other scores.

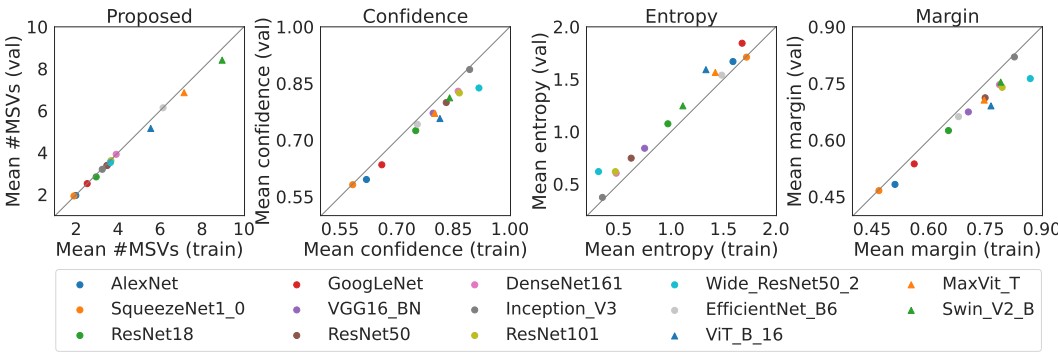

Figure 13: Comparison of scores that are estimated using the training set (x-axis) and the validation set (y-axis).

## D    EVALUATION WITH CIFAR-100 DATASET

We finetuned TorchVision's ImageNet-trained models using the CIFAR-100 training set with stochastic gradient descent for 200 epochs with a batchsize of 128. We set the weight decay to 0.0005 and the initial learning rate to 0.01, which is multiplied by 0.1 at epochs 100 and 150. For each model, we selected the best within the training epochs according to the accuracy on the CIFAR-100 validation set. We computed MSVs and other uncertainty scores for these models using 1000 images randomly sampled from the CIFAR-100 validation set. Figure 14 shows the results, in which the x-axis is the average of each score over 1000 images and the y-axis is the prediction accuracy over the entire CIFAR-100 validation set, with the rank correlations in parentheses. We can see that the average number of MSVs has a higher correlation with accuracy than other uncertainty scores.

## E    EVALUATION WITH IMAGENET-C DATASET

ImageNet-C (Hendrycks & Dietterich, 2019) consists of 75 datasets for evaluation, created by distorting images in ImageNet with different types (15 types) and levels (5 levels) of distortion. For each dataset in ImageNet-C, we computed MSVs and other uncertainty scores using randomly sampled 1000 images. We used ResNet-50 and ViT_B_16 from TorchVision as representatives of convolutional and transformer-based models, respectively. The results are shown in Figure 15, in which we can see that the rank correlation between the average number of estimated MSVs and the prediction accuracy is still as high as 0.92 in the ImageNet-C evaluation.

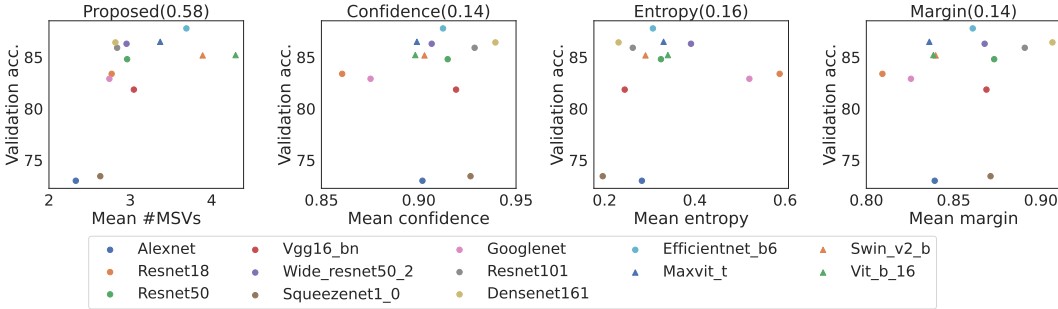

Figure 14: Average score versus top 1 accuracy for CIFAR-100 dataset. Values in parentheses for each method are rank correlations between score and accuracy.

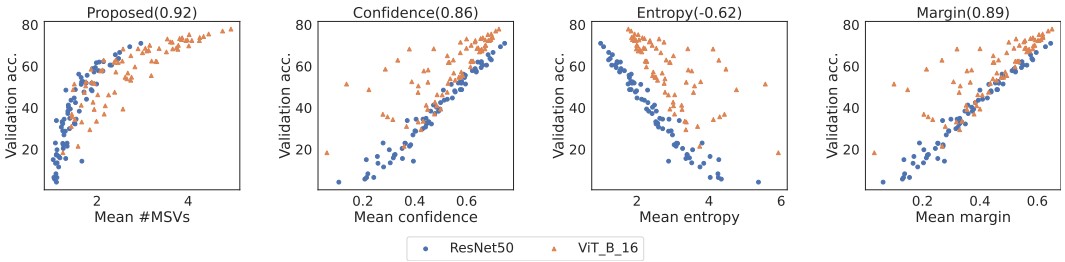

Figure 15: Average score versus top 1 accuracy for ImageNet-C dataset. Values in parentheses for each method are rank correlations between score and accuracy.

## F  APPLICATION TO A DETECTION MODEL

As a detection model, we used YOLOP (Wu et al., 2022) trained on the BDD100K dataset (Yu et al., 2020) to detect cars on the road[7]. Given a detection model and a traffic scene image $x$, we applied Algorithm 1 in the following way:

1. Detect cars in the given image using the detection model with detection threshold $\xi \in (0, 1)$, and select a detected box as the target for calculating MSVs.

2. Since YOLOP is a type of detection model that uses predefined prior boxes in its prediction, we specified the prior box to be used to detect the target.

3. Let the detection probability of the specified prior box be $p_{\text{det}}(x)$, we define $f$ used in Algorithm 1 as $f(x) = (p_{\text{det}}(x), \xi)$. Note that Sufficiency condition used in lines 8 and 14 of Algorithm 1 then means that $p_{\text{det}}(x) \geq \xi$.

Figure 8 in the main text shows an example of calculated MSVs for an image from the BDD100K validation set, where we used $\xi = 0.25$ and the red box is the target detection.

## G  ACCESSING THE EFFECT OF SPLITTING METHOD

To investigate the impact of the splitting method in the proposed method, we first conducted experiments to evaluate the prediction accuracy in terms of the number of estimated MSVs using ResNet-101 and the ImageNet validation set. We used the Voronoi partition as an additional splitting method. The results are shown in Table 3. From the table, we can see that there is a clear relationship between the accuracy and the number of estimated MSVs, despite the splitting method used in Algorithm 1.

Next, in Figure 16, we compare the MSVs obtained with SLIC or Voronoi partition using the cat images in Figures 3 and 4. Note that the estimated MSVs are shown in a single image, with each

---

[7]We used the pretrained model downloaded from `https://pytorch.org/hub/hustvl_yolop` (accessed July 27, 2023).

Table 3: Prediction accuracy for the ImageNet validation set, in which images are grouped based on the number of estimated MSVs.

| SPLIT function | #MSVs | | | | | | | | | |
|---|---|---|---|---|---|---|---|---|---|---|
| | 1 | 2 | 3 | 4 | 5 | 6 | 7 | 8 | 9 | $\geq 10$ |
| SLIC ($\beta = 16$) | 0.538 | 0.808 | 0.865 | 0.891 | 0.905 | 0.919 | 0.919 | 0.922 | 0.923 | 0.937 |
| Voronoi ($\beta = 8$) | 0.586 | 0.848 | 0.892 | 0.909 | 0.921 | 0.925 | 0.931 | 0.929 | 0.931 | 0.948 |

colored region representing one MSV. For example, the MSVs in Figure 2 are summarized as the upper left image in Figure 16. In Figure 16, we report results with $\beta = 8$ or $\beta = 16$. The first observation from Figure 16 is that the MSVs changes depending on the splitting method as well as the value of $\beta$. This is mainly due to the fact that the MSVs satisfying Definition 2 is not uniquely determined, as discussed in Section 3.2. The second but important observation from Figure 16 is that the number of estimated MSVs has a common tendency in its order given a splitting method and $\beta$ (the number of MSVs is shown in the upper part of each image). These results suggest that the average number of MSVs has a robust relationship with prediction accuracy, while the MSVs for each image should be interpreted with caution due to its dependence on several hyperparameters. A possible approach to reduce the dependence of MSVs on hyperparameters may be to estimate a common pattern from MSVs of multiple images with different hyperparameters. An example of such an approach is reported in Carter et al. (2019), where clustering methods are used to estimate common patterns.

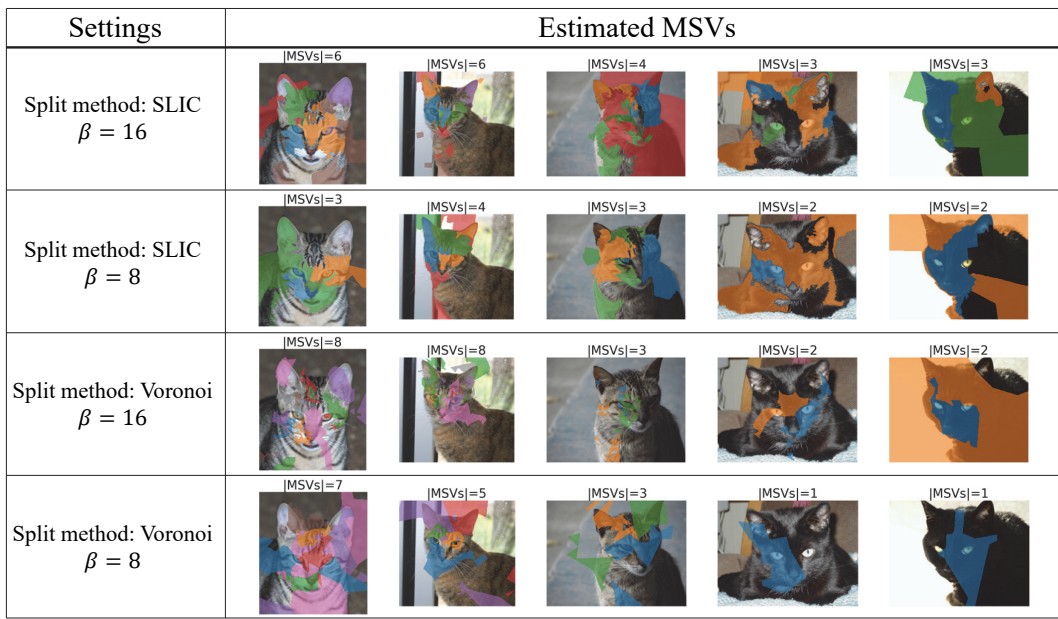

Figure 16: Comparison of estimated MSVs using different splitting methods and $\beta$. Each colored region represents an estimated MSV.

## H  ACCESSING THE EFFECT OF BASELINE VALUE

In addition to the results in Table 1, we conducted experiments where the baseline value $b$ for masking the image was white, black, or random (with a normal distribution), using the SLIC splitting and $\beta = 16$. Using ResNet-101, we computed MSVs using each baseline for images in the ImageNet validation set. Note that the label information is not used in the calculation of MSVs. We grouped the validation images according to the number of estimated MSVs and evaluated the prediction accuracy in each group; the results are shown in Table 4. From the table, we see that the accuracy of images with few MSVs (1 or 2) is relatively low, despite the baseline for masking. The accuracy

consistently increases as the number of MSVs increases when we use the average value for the baseline compared to other baseline values, especially for images with a large number of MSVs.

Table 4: Prediction accuracy for the ImageNet validation set, in which images are grouped based on the number of estimated MSVs.

| Baseline $b$ | #MSVs | | | | | | | | | |
|---|---|---|---|---|---|---|---|---|---|---|
| | 1 | 2 | 3 | 4 | 5 | 6 | 7 | 8 | 9 | $\geq 10$ |
| Average | 0.538 | 0.808 | 0.865 | 0.891 | 0.905 | 0.919 | 0.919 | 0.922 | 0.923 | 0.937 |
| White | 0.595 | 0.857 | 0.897 | 0.912 | 0.915 | 0.927 | 0.926 | 0.917 | 0.931 | 0.919 |
| Black | 0.62 | 0.854 | 0.885 | 0.897 | 0.898 | 0.911 | 0.913 | 0.903 | 0.919 | 0.914 |
| Random | 0.604 | 0.86 | 0.898 | 0.916 | 0.923 | 0.914 | 0.922 | 0.942 | 0.900 | 0.919 |

## I  SPURIOUS FEATURES FOUND AS MSVS

It is reported, for example in Carter et al. (2021), that modern neural network models can rely on spurious features to make their predictions. We can find such spurious features using MSVs, an example of which is shown in Figure 17. In Figure 17, some MSVs include only background regions, but retain the prediction of class 304 (*Leaf Beetle*).

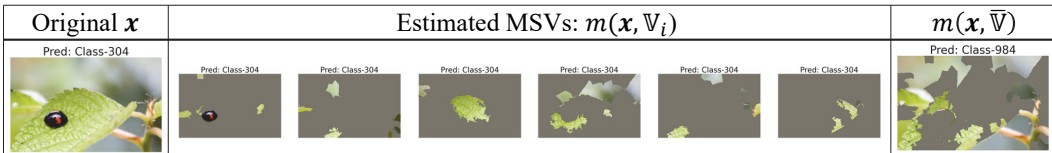

Figure 17: Spurious features found as MSVs.

## J  IMAGE URLS

The original credits for the images used in this article can be found on the websites listed below (accessed November 17, 2023). All images were used with modifications to visualize the execution results of the proposed algorithm or existing XAI methods.

- Figure 2, Figure 7, Figure 16:
    - https://www.flickr.com/photos/46785529@N00/422908475
- Figure 3:
    - https://www.flickr.com/photos/yvonne_n_1968/542453357
- Figure 4, Figure 16:
    - https://www.flickr.com/photos/24070291@N06/5760932696
    - https://www.flickr.com/photos/ke_netan_to/97744015
    - https://www.flickr.com/photos/mlspooky/208259954
    - https://www.flickr.com/photos/dharmabum1964/3317040642
- Figure 5:
    - https://www.flickr.com/photos/rusty_clark/9671631975
    - https://www.flickr.com/photos/takaokun/4342613657
    - https://www.flickr.com/photos/8143264@N08/6396168851
    - https://www.flickr.com/photos/dana_moos/14263516186
    - https://www.flickr.com/photos/llstalteri/7562636142
    - https://www.flickr.com/photos/triller/2800056925

- – https://www.flickr.com/photos/84335369@N00/3707715946
- – https://www.flickr.com/photos/naterivers/7410981534
- – https://www.flickr.com/photos/elsiehui/10354089956
- – https://www.flickr.com/photos/hulagway/6047627858
- – https://www.flickr.com/photos/nojofoto/8291625820
- – https://www.flickr.com/photos/moe/3515442082
- – https://www.flickr.com/photos/cogdog/8690615113
- – https://www.flickr.com/photos/raybouk/16051918921
- – https://www.flickr.com/photos/tomsbrain/7504267106
- – https://www.flickr.com/photos/chunso/8437950173
- – https://www.flickr.com/photos/dichohecho/4179438703
- – https://www.flickr.com/photos/shoshanah/3449328732
- – https://www.flickr.com/photos/ilove9and23/17294405089
- Figure 9:
  - – https://www.flickr.com/photos/hamed/154621901
  - – https://www.flickr.com/photos/xstuntkidx/4003674646
  - – https://www.flickr.com/photos/rhettmaxwell/2654140978
  - – https://www.flickr.com/photos/crisphotos/2472401181
- Figure 10:
  - – https://www.flickr.com/photos/tonyaustin/4008969588
  - – https://www.flickr.com/photos/picturesbyann/11115869665
  - – https://www.flickr.com/photos/jurvetson/159436192
  - – https://www.flickr.com/photos/pmunks/4663205678
- Figure 11:
  - – https://www.flickr.com/photos/9208231@N02/1120334380
  - – https://www.flickr.com/photos/kelleydenz/4529545564
  - – https://www.flickr.com/photos/psc49/8800179476
  - – https://www.flickr.com/photos/quinet/8560668127
- Figure 17:
  - – https://www.flickr.com/photos/tab2/66132838

