# OpenReview forum: "How many views does your deep neural network use for prediction?"
_ICLR.cc/2024/Conference — Submitted to ICLR 2024_

### Official Review · Reviewer_pywZ · 2023-10-27

**Soundness:** 2 fair
**Presentation:** 3 good
**Contribution:** 2 fair
**Rating:** 5
**Confidence:** 5

**Summary:**

This paper proposes Minimal Sufficient Views (MSVs), which is similar to multi-views but can be efficiently computed for real images. The proposed MSV can be used to understand the generalization ability of DNNs.

**Strengths:**

Figure 3 is vivid to illustrate the computation of the proposed MSV.

**Weaknesses:**

1. I think that the proposed MSV is a very typical and common method to evaluate the importance/attribution of each superpixel in XAI. Hence, what is the essential difference between the proposed MSV method and previous methods masking different image patches to evaluate importance/attribution.
2. Different SPLIT method will influence the final result? I think so. Hence, the proposed method indeed depends on the SPLIT method. If not, please conduct experiments for verification.
3. Will the size of view affect the final result? since in Figure 4, some msvs contain only few image region, while other contain a larger image region. Considering a msv containing more image regions often encodes more information  than a msv containing few image regions, I think msv of different numbers of image regions cannot compare fairly.

**Questions:**

Stated in Weakness.

---

> ### Author Response · Authors · 2023-11-20
> **Thank you for your review and interesting questions**
>
> Thank you for your review and interesting questions.
>
> > I think that the proposed MSV is a very typical and common method to evaluate the importance/attribution of each superpixel in XAI. Hence, what is the essential difference between the proposed MSV method and previous methods masking different image patches to evaluate importance/attribution.
>
> The essential difference between the proposed MSVs and existing XAI methods based on masking, such as RISE (Petsiuk et al., 2018), is that the proposed MSVs explicitly compute *multiple views* such that each view satisfies the sufficiency, i.e., the prediction remains unchanged from the original image.
> In contrast, RISE, for example, uses random masking to estimate the importance of each pixel (or superpixel), but the result is presented in a single heatmap representing the degree of estimated importance, without considering the notion of multiple views.
> We believe that the main contribution of our work is to show empirically that the number of views, MSVs in our context, used in the prediction has a clear relationship with the prediction accuracy.
>
>
> > Different SPLIT method will influence the final result? I think so. Hence, the proposed method indeed depends on the SPLIT method. If not, please conduct experiments for verification.
>
> To investigate the impact of the splitting method in the proposed method, we first conducted experiments to evaluate the prediction accuracy in terms of the number of estimated MSVs using ResNet-101 and the ImageNet validation set. We used the Voronoi partition as an additional splitting method. The results are shown in the table below.
> Note that we used $\beta=8$ for the Voronoi partition, mainly because we could not finish our additional experiments with $\beta=16$ with the Voronoi partition during the rebuttal period (we will add results of Voronoi with $\beta=16$ in the camera ready if accepted).
> From the table, we can see that there is a clear relationship between the accuracy and the number of estimated MSVs, despite the splitting method used in Algorithm 1.
>
> | Alg.1 Settings              | \#MSVs=1 |   2   |   3   |   4   |   5   |   6   |   7   |   8   |   9   |   10  |
> | :-------------------------- | :------: | :---: | :---: | :---: | :---: | :---: | :---: | :---: | :---: | :---: |
> | SLIC, $\beta=16$   |   0.538  | 0.808 | 0.865 | 0.891 | 0.905 | 0.919 | 0.919 | 0.922 | 0.923 | 0.937 |
> | Voronoi, $\beta=8$ |   0.586  | 0.848 | 0.892 | 0.909 | 0.921 | 0.925 | 0.931 | 0.929 | 0.931 | 0.948 |
>
> In Figure 16 of our updated manuscript, we compare the MSVs obtained with SLIC or Voronoi partition using the cat images in Figures 2 and 4.
> Note that the estimated MSVs are shown in a single image, with each colored region representing one MSV. For example, the MSVs in Figure 2 are summarized as the upper left image in Figure 16.
> In Figure 16, we report results with $\beta=8$ or $\beta=16$.
> The first observation from Figure 16 is that the MSVs changes depending on the splitting method as well as the value of $\beta$.
> This is mainly due to the fact that the MSVs satisfying Definition 2 is not uniquely determined, as discussed in Section 3.2.
> The second but important observation from Figure 16 is that the number of estimated MSVs has a common tendency in its order given a splitting method and $\beta$ (the number of MSVs is shown in the upper part of each image).
> These results suggest that the average number of MSVs has a robust relationship with prediction accuracy, while the MSVs for each image should be interpreted with caution due to its dependence on several hyperparameters.
> We think that MSVs has much room for improvement as an XAI method, but we believe that our findings on its relationship to generalization make a definite contribution.
> We have clarified these points in Appendix G of our updated manuscript.
>
>
> > Will the size of view affect the final result? since in Figure 4, some msvs contain only few image region, while other contain a larger image region. Considering a msv containing more image regions often encodes more information than a msv containing few image regions, I think msv of different numbers of image regions cannot compare fairly.
>
> First, we would like to clarify that in the proposed algorithm we do not specify the view size in advance, and the view size is determined automatically to satisfy the minimum sufficiency.
> Our results, such as Figure 1 or Table 1, suggest that the number of MSVs has a clear relationship with accuracy, even though the size of each view is not considered in these experiments.
> However, there is a possibility that we can define a better measure to predict the generalization ability of DNNs by considering not only the number of MSVs but also the size of each MSV, which we would like to leave as future work.

---

> > ### Author Response · Authors · 2023-11-23
> >
> > Dear Reviewer pywZ,
> >
> > We have submitted our response to your questions and revised our paper according to your suggestions. We sincerely appreciate your valuable feedback to improve the quality of our paper. Are there any additional questions or concerns that we can address? Thank you in advance.
> >
> > Yours, Authors

---

### Official Review · Reviewer_qNfq · 2023-10-30

**Soundness:** 2 fair
**Presentation:** 2 fair
**Contribution:** 2 fair
**Rating:** 3
**Confidence:** 4

**Summary:**

This paper proposes the concept of Minimal Sufficient Views (MSVs) as a means to understand the generalization ability of Deep Neural Networks (DNNs). The authors empirically show a relationship between the number of MSVs and prediction accuracy across various models. They argue that a multi-view perspective is crucial for understanding the generalization ability of DNNs.

**Strengths:**

The paper focuses an important and relevant topic in deep learning - the generalization ability of DNNs.

**Weaknesses:**

1.	This paper lacks a clear motivation for the proposed concept of MSVs. It is not adequately explained why MSVs are necessary or how they contribute to the understanding of generalization ability.

2.	How to use MSVs in real-world applications? MSVs need testing samples to predict the generalization ability of DNNs. However, if we can obtain testing samples, why do we need to predict, not measure, the generalization ability of DNN?

3.	Experimental results are not enough. I suggest the authors conduct experiments on NLP datasets.


4.	Lack of theoretical analysis. The authors do not theoretically explain the relationship between MSVs and the generalization ability of DNNs. Some XAI methods[cite1-4] have rigorous theoretical analysis to guarantee its faithfulness. I suggest the authors theoretical prove the faithfulness of MSVs.

[cite1] John C Harsanyi. A simplified bargaining model for the n-person cooperative game. International Economic Review, 4(2):194–220, 1963
[cite2] Lloyd S Shapley. A value for n-person games. Contributions to the Theory of Games, 2(28): 307–317, 1953.
[cite3] Michel Grabisch and Marc Roubens. An axiomatic approach to the concept of interaction among players in cooperative games. International Journal of game theory, 28(4):547–565, 1999.
[cite4] Mukund Sundararajan, Kedar Dhamdhere, and Ashish Agarwal. The shapley taylor interaction index. In International Conference on Machine Learning, pages 9259–9268. PMLR, 2020.

**Questions:**

Please see the Weaknesses.

---

> ### Author Response · Authors · 2023-11-20
> **Thank you for your review.**
>
> Thank you for your review.
>
> > This paper lacks a clear motivation for the proposed concept of MSVs. It is not adequately explained why MSVs are necessary or how they contribute to the understanding of generalization ability.
>
> As mentioned in Section 1, one of our main motivations for introducing MSVs is to investigate whether the multi-view concept, introduced by Allen-Zhu \& Li (2023) and shown to be related to the generalization ability of modern DNNs in terms of ensemble or distilled models, is also related to the generalization of (non-ensemble or distilled) DNNs.
> Our results, such as Figure 1, provide empirical evidence that the number of views, in terms of the proposed MSVs, used in the prediction is related to the generalization performance of (non-ensemble or distilled) DNNs.
>
> > How to use MSVs in real-world applications? MSVs need testing samples to predict the generalization ability of DNNs. However, if we can obtain testing samples, why do we need to predict, not measure, the generalization ability of DNN?
>
> We would like to emphasize that *labeled* test (or validation) samples are *NOT* required to calculate MSVs, and it is sufficient to prepare unlabeled test (or validation) samples.
> Furthermore, the results of our experiments, shown in Figure 13, suggest that MSVs computed using the training dataset have a high correlation with those computed using unlabeled test (or validation) samples, assuming that these datasets follow the same distribution. This suggests the possibility of selecting the best performing model using only the training dataset.
>
> > Experimental results are not enough. I suggest the authors conduct experiments on NLP datasets.
>
> Thank you for your suggestion. As described in Section 5, we think it is possible to extend the proposed MSVs to be applied to language models. However, in this paper, we focus on computer vision tasks, including classification and object detection, and leave the NLP datasets to the future work.
>
> > Lack of theoretical analysis. The authors do not theoretically explain the relationship between MSVs and the generalization ability of DNNs. Some XAI methods[cite1-4] have rigorous theoretical analysis to guarantee its faithfulness. I suggest the authors theoretical prove the faithfulness of MSVs.
>
> We agree that it is important to mathematically prove XAI methods to satisfy some axioms, such as faithfullness or sensitivity (Sundararajan et al., 2017).
> In our method, the key property (i.e., axiom) is to satisfy Minimality and Sufficiency. By satisfying these properties, all the MSVs obtained can explain the behavior of the DNN in the sense that they are the smallest regions that output a certain label by the DNN. The proposed algorithm can guarantee that each view obtained satisfies Sufficiency and $\beta$-Split-Minimality. Also $\beta$-Split Minimality becomes equivalent to Minimality of our definition when $\beta = |\mathbb{V}|$, as discussed in Section 3.2. We will add proofs of these properties in our camera-ready version if accepted.
> Regarding the relationship between MSVs and generalization performance, unfortunately only experimental evidence is available so far. However, we would like to emphasize that the relationship between the two is clear and highly correlated, as shown in Figure 1 and Table 1.

---

> > ### Author Response · Authors · 2023-11-23
> >
> > Dear Reviewer qNfq,
> >
> > We have submitted our response to your questions and revised our paper according to your suggestions. We sincerely appreciate your valuable feedback to improve the quality of our paper. Are there any additional questions or concerns that we can address? Thank you in advance.
> >
> > Yours, Authors

---

> > > ### Comment · Reviewer_qNfq · 2023-11-23
> > >
> > > I want to thank the authors for their detailed responses. However, the experimental verification and theoretical analysis of MSVs are insufficient. I strongly suggest the authors conduct experiments on other types of data, e.g., NLP datasets. Therefore, I will keep my score.

---

### Official Review · Reviewer_xok8 · 2023-10-31

**Soundness:** 2 fair
**Presentation:** 3 good
**Contribution:** 2 fair
**Rating:** 6
**Confidence:** 3

**Summary:**

This study defines the notion of minimal sufficient views (MSVs) as certain subsets of superpixels that preserves the prediction result of the DNN, inspired by the multi-views introduced in Allen-Zhu & Li (2023). This study visualizes MSVs on several examples and empirically discovers that the number of MSVs is positively correlated to the prediction accuracy of the model, thus providing a new perspective for understanding the generalization ability of DNNs.

**Strengths:**

1.	The mathematical definition of MSVs and the greedy algorithm for computing MVSs are both clearly written. I appreciate the execution example in Figure 3 which makes the computing process intuitive.
2.	The paper is easy to read and follow.
3.	The number of MSVs provides a novel view for estimating and comparing the generalization ability of DNNs.

**Weaknesses:**

1.	The notion of MSVs in this paper is quite similar to the Sufficient Input Subsets (SIS) proposed in [cite 1, cite 2], which is expected to be discussed in the Related Work section. SIS characterizes the minimal subset of input pixels (pixels outside of this subset is masked) for the model to achieve a certain level of confidence score. In this way, the proposed Minimal Sufficient Views seem to be a simple extension of the Sufficient Input Subsets, so the authors are encouraged to clarify the differences between the two methods.
2.	The previous work [cite 2] has noted an interesting phenomenon: for many images in CIFAR-10 and ImageNet, the size of the Sufficient Input Subsets (SIS) is quite small (e.g., only 5% to 10% of total number of pixels) and pixels in SIS are sometimes located outside of the target object. This means that the model might learn shortcut solutions, such as using blue pixels within the sky region to predict the bird class. Since the definition of MSV is similar to SIS, I wonder if a similar phenomenon occurs in this paper. From the current figures presented in this paper, most MSVs are located on the target object and seem to have clear semantic meanings, but I’m not sure if there are some “failure cases” in which the MSVs corresponds to patterns that are not related to the target object (e.g., pixels within the sky region or the grass region).
3.	About the baseline value for masking the image. Although using the average value of the pixels in the training data as the baseline value is a common practice in literature, it is encouraged to test if the derived MSVs and the relationship to the generalization ability are robust under different choices of baseline values. This is because in most views (a masked image), the size of the mask is quite large, thus greatly influencing the output of the model. It is not clear if the current conclusions still hold under a different baseline value.
4.	I do not quite agree with the claim that “MSVs with common features were obtained for multiple images” in the same class on Page 6. The notion of “left eye”, “right eye” are based on human perception, but it is not clear whether the model also encodes these features for inference. Moreover, the MSVs are defined in the pixel space instead of the feature space. It is not appropriate to simply claim that feature “a left eye with circular shape on a black cat” is equivalent to the feature “a left eye with an almond shape on a white cat”.

[cite 1] Carter, Brandon, et al. What made you do this? understanding black-box decisions with sufficient input subsets. International Conference on Artificial Intelligence and Statistics, 2019.

[cite 2] Carter, Brandon, et al. Overinterpretation reveals image classification model pathologies. Advances in Neural Information Processing Systems, 2021.

**Questions:**

1.	I wonder how will different superpixel methods, such as SLIC and the Voronoi partition, influence the resulted MSVs for the same input image. Will the result be similar or totally different?
2.	Minor. The visualization result of GradCAM in Figure 7 is a bit weird. It is suggested to check the original GradCAM paper and compare this result with that of the original paper.

---

> ### Author Response · Authors · 2023-11-20
> **Responses (1/3)**
>
> Thank you for your insightful review.
>
> > The notion of MSVs in this paper is quite similar to the Sufficient Input Subsets (SIS) proposed in [cite 1, cite 2], which is expected to be discussed in the Related Work section. SIS characterizes the minimal subset of input pixels (pixels outside of this subset is masked) for the model to achieve a certain level of confidence score. In this way, the proposed Minimal Sufficient Views seem to be a simple extension of the Sufficient Input Subsets, so the authors are encouraged to clarify the differences between the two methods.
>
> Thank you for pointing us to the SIS references.
> We agree that the definition of SIS is similar to our MSVs, but we think there are several critical differences between the two, as follows:
> * SIS papers [cite 1, cite 2] do not discuss the relationship between the number of the estimated SIS and the prediction accuracy.
> In contrast, we are the first to empirically demonstrate the relationship between the number of views, in terms of MSVs, and accuracy, which we believe is the main contribution of our paper.
> * As the reviewer pointed out, the definition of sufficiency is slightly different between SIS and MSVs (SIS is based on the confidence score and MSVs is based on the prediction class).
> We note that the definition of minimality is also different in SIS and MSVs, which is more essential because this difference leads to the difference in the algorithmic design of the two, resulting in the efficient computation of MSVs.
> Minimality in SIS is defined as "no subset of $x$ satisfies the sufficiency", while minimality of MSVs is given by "$x$ does not satisfy the sufficiency when an element is removed from $x$", meaning that the minimality condition in SIS is stronger than that in MSVs.
> As stated in [cite 1], in order to satisfy minimality in SIS, the BackSelect procedure in SIS does not terminate immediately when sufficiency is violated.
> In contrast, the proposed algorithm terminates the EstimateMSV procedure as soon as sufficiency is violated, resulting in a more efficient computation of MSVs.
> We also propose $\beta$-split minimality to further accelerate the computation of MSVs. The table below shows the computation time required to compute MSVs with respect to $\beta$, which is the average time to obtain MSVs for a single image.
>
> | $\beta$                      |   4  |   8  |  16  |   32  |
> | ---------------------------- | :--: | :--: | :--: | :---: |
> | Average Time per Image (sec) | 0.46 | 1.97 | 8.48 | 41.56 |
>
> * Batched Gradient SIS is proposed in [cite 2] to improve the computational efficiency of the original SIS.
> The advantage of the proposed algorithm over Batched Gradient SIS is that the proposed algorithm does not require gradient computation, allowing it to be applied to black-box models.
>
> We have added a discussion of these points in Section 2 of the updated manuscript.
>
> > The previous work [cite 2] has noted an interesting phenomenon: for many images in CIFAR-10 and ImageNet, the size of the Sufficient Input Subsets (SIS) is quite small (e.g., only 5% to 10% of total number of pixels) and pixels in SIS are sometimes located outside of the target object. This means that the model might learn shortcut solutions, such as using blue pixels within the sky region to predict the bird class. Since the definition of MSV is similar to SIS, I wonder if a similar phenomenon occurs in this paper. From the current figures presented in this paper, most MSVs are located on the target object and seem to have clear semantic meanings, but I’m not sure if there are some “failure cases” in which the MSVs corresponds to patterns that are not related to the target object (e.g., pixels within the sky region or the grass region).
>
> We examined the estimated MSVs with ResNet-101 and images from OpenImages, and found an example where the regions outside the class object are specified as MSVs.
> The results are shown in Figure 17 of our updated manuscript, where the image is classified as class 304 (*Leaf Beetle*), while some MSVs contain only background leaf regions.
> From the definition of the MSVs, we can see in Figure 17 that the ResNet-101 model has multiple views to predict the image as a leaf beetle, some of which include only the background leaf region.

---

> > ### Author Response · Authors · 2023-11-20
> > **Responses (2/3)**
> >
> > > About the baseline value for masking the image. Although using the average value of the pixels in the training data as the baseline value is a common practice in literature, it is encouraged to test if the derived MSVs and the relationship to the generalization ability are robust under different choices of baseline values. This is because in most views (a masked image), the size of the mask is quite large, thus greatly influencing the output of the model. It is not clear if the current conclusions still hold under a different baseline value.
> >
> > We conducted additional experiments where the baseline value for masking the image was white, black, or random (with a normal distribution), using the SLIC splitting and $\beta=16$.
> > The rest of the experimental setup is the same as in Table 1, i.e., we computed MSVs with each baseline for images in the ImageNet validation set using ResNet-101.
> > We grouped the validation images according to the number of estimated MSVs and evaluated the
> > prediction accuracy in each group; the results are shown in the table below.
> >
> > | Alg.1 Settings              | \#MSVs=1 |   2   |   3   |   4   |   5   |   6   |   7   |   8   |   9   |   10  |
> > | :-------------------------- | :------: | :---: | :---: | :---: | :---: | :---: | :---: | :---: | :---: | :---: |
> > | SLIC, $\beta=16$, Average   |   0.538  | 0.808 | 0.865 | 0.891 | 0.905 | 0.919 | 0.919 | 0.922 | 0.923 | 0.937 |
> > | SLIC, $\beta=16$, White     |   0.595  | 0.857 | 0.897 | 0.912 | 0.915 | 0.927 | 0.926 | 0.917 | 0.931 | 0.919 |
> > | SLIC, $\beta=16$, Black     |   0.62   | 0.854 | 0.885 | 0.897 | 0.898 | 0.911 | 0.913 | 0.903 | 0.919 | 0.914 |
> > | SLIC, $\beta=16$, Random    |   0.604  |  0.86 | 0.898 | 0.916 | 0.923 | 0.914 | 0.922 | 0.942 | 0.900 | 0.919 |
> >
> > We see that the accuracy of images with few MSVs (1 or 2) is relatively low, despite the baseline for masking.
> > The accuracy consistently increases as the number of MSVs increases when we use the average value for the baseline compared to other baseline values, especially for images with a large number of MSVs.
> > We have included these results in Appendix H of our updated manuscript.
> >
> > > I do not quite agree with the claim that “MSVs with common features were obtained for multiple images” in the same class on Page 6. The notion of “left eye”, “right eye” are based on human perception, but it is not clear whether the model also encodes these features for inference. Moreover, the MSVs are defined in the pixel space instead of the feature space. It is not appropriate to simply claim that feature “a left eye with circular shape on a black cat” is equivalent to the feature “a left eye with an almond shape on a white cat”.
> >
> > We agree that the MSVs are defined in input (pixel) space, not feature space.
> > We also agree that it is not clear from our results whether the model encodes features such as "left eye" or "right eye".
> > What we can clearly say from Figure 4 is that the masked images are all classified as Class 285 (*Egyptian Cat*) by the prediction model (here ResNet-101), as well as the original images.
> > As the reviewer pointed out, the definition of "common feature" here is not mathematically clear, and we used the term based on our (authors') perception.
> > We think that a mechanical way to infer (well-defined) common features from MSVs of multiple images is a future research direction. An example of such an extension is reported in the SIS paper [cite 1], where clustering methods are used to estimate common patterns.
> > We have added a discussion of these points in Appendix G of the updated manuscript.

---

> > > ### Author Response · Authors · 2023-11-20
> > > **Responses (3/3)**
> > >
> > > > I wonder how will different superpixel methods, such as SLIC and the Voronoi partition, influence the resulted MSVs for the same input image. Will the result be similar or totally different?
> > >
> > > To investigate the impact of the splitting method in the proposed method, we first conducted experiments to evaluate the prediction accuracy in terms of the number of estimated MSVs using ResNet-101 and the ImageNet validation set. We used the Voronoi partition as an additional splitting method. The results are shown in the table below.
> > > Note that we used the average as the baseline and $\beta=8$ for the Voronoi partition, mainly because we could not finish our additional experiments with $\beta=16$ with the Voronoi partition during the rebuttal period (we will add results of Voronoi with $\beta=16$ in the camera ready if accepted).
> > > From the table, we can see that there is a clear relationship between the accuracy and the number of estimated MSVs, despite the splitting method used in Algorithm 1.
> > >
> > > | Alg.1 Settings              | \#MSVs=1 |   2   |   3   |   4   |   5   |   6   |   7   |   8   |   9   |   10  |
> > > | :-------------------------- | :------: | :---: | :---: | :---: | :---: | :---: | :---: | :---: | :---: | :---: |
> > > | SLIC, $\beta=16$, Average   |   0.538  | 0.808 | 0.865 | 0.891 | 0.905 | 0.919 | 0.919 | 0.922 | 0.923 | 0.937 |
> > > | Voronoi, $\beta=8$, Average |   0.586  | 0.848 | 0.892 | 0.909 | 0.921 | 0.925 | 0.931 | 0.929 | 0.931 | 0.948 |
> > >
> > > In Figure 16 of our updated manuscript, we compare the MSVs obtained with SLIC or Voronoi partition using the cat images in Figures 2 and 4.
> > > Note that the estimated MSVs are shown in a single image, with each colored region representing one MSV. For example, the MSVs in Figure 2 are summarized as the upper left image in Figure 16.
> > > In Figure 16, we report results with $\beta=8$ or $\beta=16$.
> > > The first observation from Figure 16 is that the MSVs changes depending on the splitting method as well as the value of $\beta$.
> > > This is mainly due to the fact that the MSVs satisfying Definition 2 is not uniquely determined, as discussed in Section 3.2.
> > > The second but important observation from Figure 16 is that the number of estimated MSVs has a common tendency in its order given a splitting method and $\beta$ (the number of MSVs is shown in the upper part of each image).
> > > These results suggest that the average number of MSVs has a robust relationship with prediction accuracy, while the MSVs for each image should be interpreted with caution due to its dependence on several hyperparameters.
> > > We think that MSVs has much room for improvement as an XAI method, but we believe that our findings on its relationship to generalization make a definite contribution.
> > > We have clarified these points in Appendix G of our updated manuscript.
> > >
> > > > Minor. The visualization result of GradCAM in Figure 7 is a bit weird. It is suggested to check the original GradCAM paper and compare this result with that of the original paper.
> > >
> > > We replaced the GradCAM result with the one applied to the **last** layer of the 4th residual layer of ResNet-101, instead of the **first** layer of the 4th residual layer. See Figure 7 of our updated manuscript and footnote 6.

---

> > > > ### Author Response · Authors · 2023-11-23
> > > >
> > > > Dear Reviewer xok8,
> > > >
> > > > We have submitted our response to your questions and revised our paper according to your suggestions. We sincerely appreciate your valuable feedback to improve the quality of our paper. Are there any additional questions or concerns that we can address? Thank you in advance.
> > > >
> > > > Yours, Authors

---

> > > ### Comment · Reviewer_xok8 · 2023-11-23
> > >
> > > I would like to thank the authors for their detailed responses. I appreciate the added experiments and discussions, which address most of my concerns. I have raised my score accordingly.
> > >
> > > One concern remains for the results under different settings of the baseline value. In the 2nd part of the response, a table shows the accuracy on images with different numbers of MSVs, under different baseline value settings. The validation accuracy shows a monotonically increasing trend with the number of MSVs when the baseline value is set to the average pixel value. However, the monotonicity breaks for other settings of baseline values (although from #MSVs=1 to #MSVs=5, the monotonicity holds for all baseline values). I would appreciate it if the authors could add some discussions on this issue (maybe in a future version of the paper).

---

> > > > ### Author Response · Authors · 2023-11-23
> > > >
> > > > Thank you for your valuable suggestions.
> > > >
> > > > > One concern remains for the results under different settings of the baseline value. In the 2nd part of the response, a table shows the accuracy on images with different numbers of MSVs, under different baseline value settings. The validation accuracy shows a monotonically increasing trend with the number of MSVs when the baseline value is set to the average pixel value. However, the monotonicity breaks for other settings of baseline values (although from #MSVs=1 to #MSVs=5, the monotonicity holds for all baseline values). I would appreciate it if the authors could add some discussions on this issue (maybe in a future version of the paper).
> > > >
> > > > A possible reason why black or white baselines do not work well can be explained as follows: When we use black as a baseline, an image with a majority of its color near black tends to have a larger number of MSVs compared to an image with a majority of its color near white.
> > > > This is because the near-black image can easily maintain its prediction when the black baseline is used.
> > > > The opposite is also true (white images tend to have many MSVs when the white baseline is used).
> > > > Note, however, that these reasons do not explain the random baseline case well.
> > > >
> > > > Another possible reason for the monotonicity breaks for #MSVs > 6 is that the number of images with #MSVs > 6 is smaller compared to those with #MSVs <= 5. Therefore, the prediction accuracy for #MSVs > 6 has a larger estimation variance, which can lead to the monotonicity breaks.
> > > >
> > > > We will include a discussion of these points in Appendix H of the final manuscript.

---

> > > > > ### Comment · Reviewer_xok8 · 2023-11-23
> > > > >
> > > > > Thank you for the timely feedback. I'd be happy to see such discussions in a future edit of the paper. I will keep my current score of 6.

---

### Official Review · Reviewer_XK9X · 2023-10-31

**Soundness:** 2 fair
**Presentation:** 3 good
**Contribution:** 2 fair
**Rating:** 5
**Confidence:** 4

**Summary:**

This paper proposed MSV (Minimal Sufficient View) -- a method to get the parts(or features) of an input (sample/vector) that are minimal and sufficient to preserve a model's classification decision. Given a model and sample, the method outputs the MSV for that specific sample. The experiments conducted shows a strong correlation between MSV and accuracy (if model uses more MSV, it tends to have a higher accuracy). The authors also give a comparison between MSV and previous XAI methods.

**Strengths:**

- The paper states an important idea that a model's prediction relies on multiple features/views, and the experimental result (Table 1) provides a strong evidence
- I find the finding in Table 1 interesting (previous point)
- I like the idea that this method can be used to select model without label
- I find the visualizations to be very helpful in understanding the idea

**Weaknesses:**

- The method sounds computationally expensive. Given that the author pitch this as a model selection/XAI method, an analysis on runtime will help
- Although I find the multi view idea and its relation to accuracy interesting, I find the method lack coherence. Is it an XAI method, or a model selection method?
- In either case, the evaluation is lacking. Not sufficient comparison to existing XAI/model selection methods.
- What is the difference between single view based method (like gradcam) with combining all MSV into a single image?
- Minor, but in Definition 1, c(f(x)): c has not been defined before.

Overall, I am not very opposed to accept this paper, as long as the author can give a convincing argument where does this method lies (is it an XAI method/model selection? or none of the two? if the latter, how is this idea significant?). I am not strongly opposed to rejecting this paper either.

**Questions:**

see weakness

---

> ### Author Response · Authors · 2023-11-20
> **Thank you for your review and interesting questions.**
>
> Thank you for your review and interesting questions.
>
> > The method sounds computationally expensive. Given that the author pitch this as a model selection/XAI method, an analysis on runtime will help
>
> We evaluated the computation time of Algorithm 1 with ResNet-101 using 1000 images sampled from the ImageNet validation set used in the evaluation of Figure 1.
> The experiment was performed on a single machine with a single GPU (Nvidia A100).
> The results are summarized in the table below, in which the computation time is averaged over 1000 images.
> We can see that the computation time decreases as we use smaller $\beta$.
> From the table, it will take about less than 8 minutes to compute the MSVs for these 1000 images when $\beta=4$.
> Note that the average number of MSVs still has a high correlation with the generalization ability when $\beta=4$, as shown in Figure 12.
> Since MSVs can be computed in parallel with respect to each image, we can further accelerate the computation of MSVs by applying parallel computing, although the results in the table below are computed on a single thread in a single machine.
> We have included these results in Appendix B.1 of our updated manuscript.
>
> | $\beta$                      |   4  |   8  |  16  |   32  |
> | ---------------------------- | :--: | :--: | :--: | :---: |
> | Average Time per Image (sec) | 0.46 | 1.97 | 8.48 | 41.56 |
>
>
> > Although I find the multi view idea and its relation to accuracy interesting, I find the method lack coherence. Is it an XAI method, or a model selection method?
>
> We believe that the main contribution of our paper is to first show the relationship between the number of views used in the prediction of DNN models and their generalization ability, where views in a prediction are defined in a simple way, which we call MSVs, and an algorithm is proposed to (approximately) compute them efficiently.
> In this sense, the main application of the proposed algorithm is the selection of DNN models without the need of a labeled holdout set.
>
> Under the observation that the multi-view perspective is related to the performance of DNN models, our secondary claim is that XAI methods, especially attribution visualization, should take multi-view into account.
> The difference between MSVs as an XAI method and many existing XAI methods is that MSVs provides multiple sets of sufficient features. DNNs determine their output by combining multiple features, and there may be multiple combinations of these features.
> Existing methods that present a single heat map cannot represent such a combinatorial aspect of DNNs.
> In contrast, each of the MSVs presented in the proposed method corresponds to a region that is sufficient and minimal to maintain the output of the DNN. This allows for a more accurate understanding of the nonlinear behavior of the DNN.
> However, we think that MSVs has a lot of room for improvement to be used as an XAI method.
> For example, the dependence of the estimated MSVs on the splitting method used should be reduced to be used for XAI.
>
> > In either case, the evaluation is lacking. Not sufficient comparison to existing XAI/model selection methods.
>
> For evaluation as a model selection method, we compared the proposed method with baseline methods, including average confidence, average entropy, and average margin, as shown in Figure 6.
> These methods are relatively primitive, but we believe that these baselines are reasonable choices because, for example, average confidence has a high correlation with the test accuracy when considering only CNN models.
> We agree that more sophisticated methods should be better included in the comparison, but we have had difficulty finding methods that can be applied to DNN model selection using only an unlabeled validation set or the training set itself.
> We would appreciate having other candidates for comparison.
>
> > What is the difference between single view based method (like gradcam) with combining all MSV into a single image?
>
> We have added Figure 16 in the updated manuscript, which shows the estimated MSVs in a single image.
> For example, the MSVs in Figure 2 are summarized as the upper-left image in Figure 16.
> Single-view methods such as GradCAM show attribute importance by the intensity of a single color. In contrast, MSVs use multiple colors to represent different views, each of which is sufficient to maintain the prediction.
>
> > Minor, but in Definition 1, c(f(x)): c has not been defined before.
>
> The definition of $c_f(x)$ is given in the first paragraph of Section 3.2.

---

> > ### Author Response · Authors · 2023-11-23
> > **Official Comment by Authors**
> >
> > Dear Reviewer XK9X,
> >
> > We have submitted our response to your questions and revised our paper according to your suggestions. We sincerely appreciate your valuable feedback to improve the quality of our paper.
> > Are there any additional questions or concerns that we can address? Thank you in advance.
> >
> > Yours,
> > Authors

---

### Meta-Review · Area_Chair_8eLc · 2023-12-10

**Metareview:**

This paper proposes Minimal Sufficient Views (MSVs), which is similar to multi-views but can be efficiently computed for real images. MSVs are minimal and sufficient to preserve a model's prediction.  The paper shows that DNN model uses multiple views in its prediction, in the sense of MSVs mentioned above. The authors also provide empirical evidence to show that number of views used in a prediction depends on both the model and input. Furthermore, there is a clear relationship between the number of views and the prediction accuracy.

**Justification For Why Not Higher Score:**

I agree with the reviewers that the novelty is somewhat limited and both theoretical & empirical aspects of the paper need improvement. I recommend the authors to address these concerns.

**Justification For Why Not Lower Score:**

N/A

---

### Decision · Program_Chairs · 2024-01-16

Reject